

# An efficient networking solution for extending and controlling wireless sensor networks using low-energy technologies

Mostafa Ibrahim Labib[1], Mohamed ElGazzar[2], Atef Ghalwash[1] and Sarah Nabil AbdulKader[1,3]

[1] Computer Science, Faculty of Computers and Artificial Intelligence, Helwan University, Cairo, Egypt
[2] Vodafone Company, Cairo, Egypt
[3] Computer Science, Faculty of Computer Studies, Arab Open University, Cairo, Egypt

## ABSTRACT

Wireless sensor networks connect a set of highly flexible wireless devices with small weight and size. They are used to monitor and control the environment by organizing the acquired data at a central device. Constructing fully connected networks using low power consumption sensors, devices, and protocols is one of the main challenges facing wireless sensor networks, especially in places where it is difficult to establish wireless networks in a normal way, such as military areas, archaeological sites, agricultural districts, construction sites, and so on. This paper proposes an approach for constructing and extending Bi-Directional mesh networks using low power consumption technologies inside various indoors and outdoors architectures called "an adaptable Spider-Mesh topology". The use of ESP-NOW protocol as a communication technology added an advantage of longer communication distance versus a slight increase of consumed power. It provides 15 times longer distance compared to BLE protocol while consuming only twice as much power. Therefore, according to our theoretical and experimental comparisons, the proposed approach could provide higher network coverage while maintaining an acceptable level of power consumption.

# INTRODUCTION

Monitoring, measuring and controlling systems are rapidly evolving in today's world. Embedded systems, the Internet of Things (IoT), remote environmental monitoring, and smart home automation are just a few of the applications that have motivated by the creation of Wireless Sensor Networks (WSNs) (*Labib et al., 2019*). Wireless sensor networks typically connect smart sensors to other IoT modules for collecting, monitoring, and remotely controlling real-time physical parameters such as air temperature, humidity, air pressure, soil moisture, and other environmental variables. The collected data is sent to a central base station for automatic decision-making or notification of decision-makers (*Babiuch, Foltýnek & Smutný, 2019*).

Corresponding authors
Mostafa Ibrahim Labib,
mostafa_ebrahim87@yahoo.com
Sarah Nabil AbdulKader,
nabil.sarah@gmail.com

Recently, the promising use of WSNs and IoT-based systems has prompted researchers to build wireless networks in places where it is difficult to do so using conventional low-power technologies, such as military zones, marine environment (*Xu, Shen & Wang, 2014*) construction sites (*Nguyen, Nguyen & Ha, 2020*), open-air archaeological sites (*Perles et al., 2018*), agriculture districts (*Kour & Arora, 2020*) and others.

Implementing low-power consumption, low-cost, and high-flexibility wireless devices with small weight and size is required to build and scale up WSNs and IoT applications. Due to its technical specifications, performance properties, functionality, and affordability, the Esp8266 and Esp32 are two excellent low power consumption solutions for WSNs and IoT-based systems (*Hoddie & Prader, 2020*).

The Bluetooth Low Energy (BLE) and the ESP-NOW protocols are two low-power consumption protocols that can be used to configure Wireless Sensor Networks (WSNs) on ESP32 devices. The standard BLE protocol is limited to a star topology, as well as a Master/Slave connection methodology between network nodes; moreover, a short coverage distance between network nodes (*Labib et al., 2019*).

The ESP-NOW protocol is a powerful proprietary protocol for establishing a Wireless Senor Networks with mesh topology. The Sender/Receiver connection methodology between the network nodes allows each node to operate as a transmitter, receiver, or transceiver.

This paper proposes a generic approach for constructing a fully connected network using low power consumption technology in places where it is difficult to establish a network using traditional technologies. This approach leveraging the advantages of employing an adaptive spider mesh topology using the ESP-NOW protocol to connect nodes; moreover this paper studies two important factor in constructing the desired network: the maximum distance between network nodes within various indoor and outdoor architectures, as well as the maximum network nodes lifetime using ESP-NOW and Bluetooth low Energy protocols.

The remainder of this paper is organized as follows: "Related work" reviews an overview of current wireless communication technologies. "Wireless sensor networks communication protocols" discusses the major characteristics of the two main communication protocols used in this paper "BLE and ESP-NOW protocols". "An adaptive spider - mesh topology" presents the main characteristics of the proposed approach "an adaptive Spider - Mesh topology". "Experimental results" analyzes the experimental results of the proposed approach. Finally, "Conclusion and future work" has the conclusions and possible future works.

## RELATED WORK

A number of criteria must be considered while building a reliable, stable, and scalable wireless sensor network and IoT-based applications; including communication technologies, supported network topologies, communication methodologies, power consumption, limited bandwidth, and environmental constraints.

Radio waves have recently been employed to connect network devices, which allowed greater flexibility when installing multiple devices in the desired locations. Wireless

**Table 1  Wireless communication technologies comparison (adapted from *Abdulhussien & Ibrahim, 2020*).**

| Specification | Bluetooth | Z-Wave | ZigBee | Thread |
|---|---|---|---|---|
| Network type | Point-to-point, scatternet | Mesh | Mesh | Mesh |
| Maximum nodes connected | 7 | 232 | 65,536 | 250 |
| Distance | Approximately 10–100 m | 100 m with no obstructions | Approximately 10–20 m | Normally 20–30 m |
| Throughput | 24 Mbit/s | 40 kbps | 110 kbps maximum | 250 kbps |
| Spread spectrum | AFH | DSSS | DSSS | DSSS |
| Modulation | GFSK | GFSK | OQPSK | OQPSK |
| Data | Exchanging data | Monitoring and control data | Monitoring and control data | Monitoring and control data |
| Power consumption | Low | Low | Low | Low |
| Voice capable | Yes | Yes | Yes | No |
| Security | 56–128 bit key derivation | AES-128 | AES-128 | Banking-class, public-key cryptography |
| Cost | Low | High | Low | Low |
| Backwards compatibility | Yes | Yes | No | Yes |

communication technologies are essential not only because they potentially replace wired networks , but they also enable peoples to use, control and monitor their devices everywhere in the world by connecting them to the Internet (*Reinisch et al., 2007*).

Bluetooth, Z-wave, ZigBee, and Thread are some of the most widely used low-power wireless communication technologies for connecting devices and transferring/controlling data. The most common specs for these technologies are listed in Table 1 (*Morales, Parado & Pasaoa, 2021*). These technologies supports Star, Tree and Mesh topologies. The Mesh topology is the best option for constructing, extending and controlling the Wireless Sensor Networks (WSNs).

Researchers have studied and evaluated the effects of star and mesh Wireless Sensor Network (WSNs) topologies on response time, throughput, traffic drop and delay using Zigbee communication protocols. The results demonstrated that the network functions better with mesh topology than star topology (*Abdulhussien & Ibrahim, 2020*).

Other research measured and analyzed the average value of delay, throughput and packet loss parameters of star mesh and tress topologies using Zigbee communication protocols. The result show that the start topology is stable on measuring throughput and packet loss besides; it had the smallest delay value. However the mesh and tree topologies had the advantage of being able to send data over longer distances and adding more nodes than the star topology (*Soijoyo & Ashari, 2017*).

Wireless sensor networks (WSNs) with high power consumption technologies can extend network lifetime and enable efficient, reliable, and dependable wireless

communications (*Yick, Mukherjee & Ghosal, 2008*). The power consumption technologies is divided into three parts; Sensing, Communication and Data processing. Bluetooth Low Energy and ESP-NOW protocols are two ultra-low-power consumption communication protocols that can be used in a wide variety of WSNs and IoT based systems.

According to the Bluetooth Low Energy Core Specification, Bluetooth Low Energy is used to establish three kinds of network structure: Star, Mesh and Tree structure network (*Li, Zhang & Marie, 2019*). Star topology is the simplest topology. BLE mesh is the most flexible and reliable network structure which has the ability to extend the network coverage area. It is a complicated and is not considered a power consumption and latency efficient protocol (*Ghori, Wan & Sodhy, 2020*). A Tree structure network consists of three different types of nodes: the root node, the intermediate node and the leaf node. it can connect more nodes than the star network. Moreover, its routing rules are significantly simpler than mesh routing rules (*Li, Zhang & Marie, 2019*).

The goal of this article is constructing an efficient network with mesh topology that can be used to extend and control Wireless Sensor Networks using low-energy communication protocols, as well as studying and analyzing the permissible distance between network nodes and the consumed power of each node. Selecting an energy-efficient routing mechanism for our suggested approach is an challenging task that necessitates a lot of experiments and comparisons. As a result, the extended work of this paper will focus heavily on choosing the most appropriate routing protocol for the network topology and the communication protocol that will be presented in this paper.

## WIRELESS SENSOR NETWORKS COMMUNICATION PROTOCOLS

Communication protocols are essential for connecting devices and sharing data in wireless sensor networks and Internet of Things devices. Specific communication protocols are required for building networks for monitoring and controlling construction sites, the marine environment, and archaeological sites. These protocols should be low-power and capable of sharing data across all network nodes. The following communication protocols will be used to test the proposed approach in this study.

### Bluetooth low energy

Classic Bluetooth, Bluetooth Low Energy, Wi-Fi, and ESP-NOW Protocols are all supported by the ESP32 boards. Compared to Classic Bluetooth, Bluetooth Low Energy is designed to use significantly less power while maintaining a similar communication range.

To connect devices using the BLE protocol, devices can act as a Central/Master (smart phones or PCs) or Peripheral/Slave (small devices such as smart watches or ESP32 boards). Peripheral devices advertise their existence and wait for the central device to connect to them, whereas the central device scans nearby devices and connects them. Devices can be either a Client or a Server after establishing a BLE connection, as depicted in Fig. 1. A server

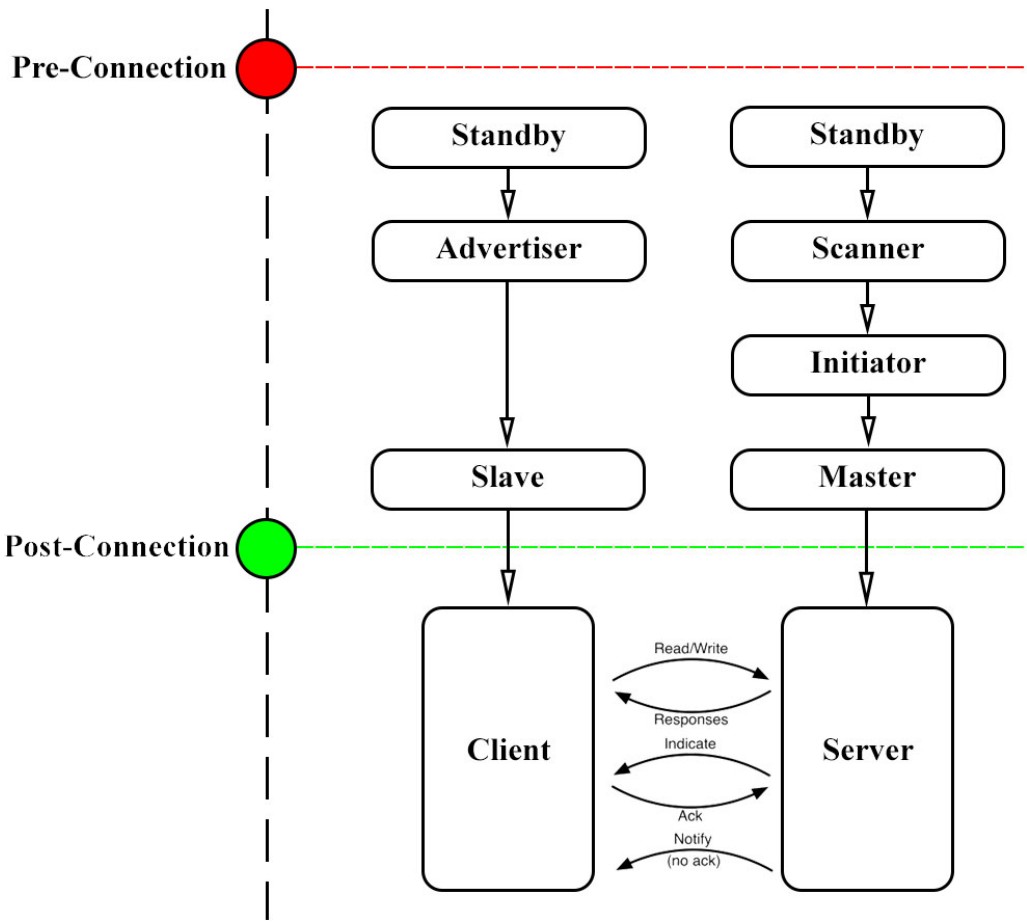

**Figure 1** **Point-to-point connection methodology using BLE protocol.**

device has local resources such as "profiles, services, and characteristics" that clients can read (*Santos & Santos, 2020*).

Recently, mesh networks can be configured utilizing the BLE protocol, although there are several limitations. According to our experiments, the maximum distance between the server and clients is six meters, and the maximum number of clients connected to the server at the same time is three devices.

Researchers have been able to overcome these challenges in a variety of ways, including switching some client devices into break mode to allow other devices to connect in their place, or using a time division system to switch Server/Client mode for some devices on the network, but these techniques are extremely complex (*Patti, Leonardi & Bello, 2016*). The existing limitations of using the BLE protocol for establishing and expanding the Wireless Sensors Network pushed us to look for other low power consumption protocols that could address these difficulties.

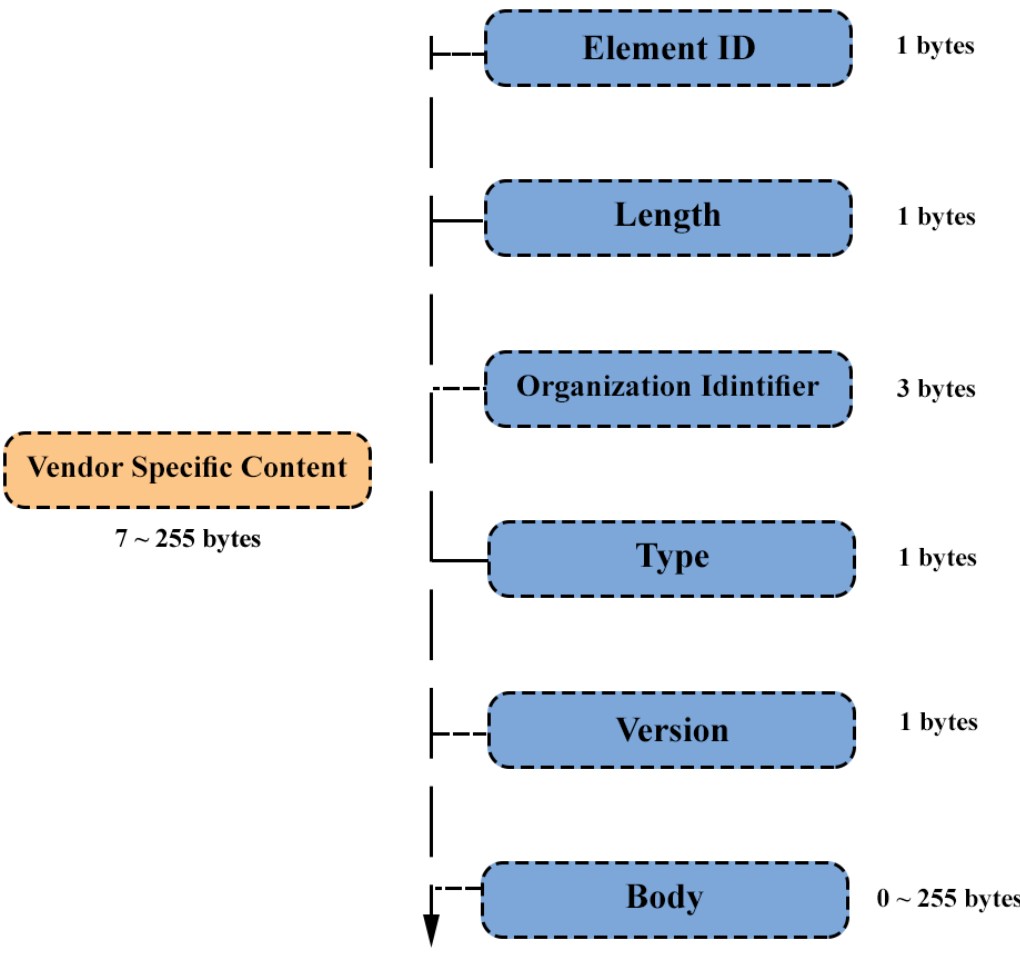

**Figure 2** ESP-NOW vendor-specific frame format (adapted from *ESP-IDF (2020)*).

## ESP-NOW

ESP-NOW is a fast wireless communication proprietary protocol developed by "Espressif organization" that may be used to transfer small messages (up to 250 bytes) between ESP32 boards (https://www.espressif.com/en/products/software/esp-now/overview). As demonstrated in Fig. 2, data is encapsulated in a vendor-specific action frame and then sent from one device to another. The pairing between devices is required prior to their communication. After pairing, the connection becomes secure and peer-to-peer, with no need for handshake process (*Pasic, Kuzmanov & Atanasovski, 2020*).

The ESP-NOW protocol is similar to the low-power 2.4 GHz wireless connectivity. This protocol allows multiple low-power devices to communicated to each other and exchange data between ESP32 boards without the use of Wi-Fi or Bluetooth technologies (*Random Nerd Tutorials, 2020a*).

Table 2 summarizes the essential differences between the Bluetooth Low Energy and ESP-NOW protocols. As demonstrated in Fig. 3, the ESP-NOW protocol allows to configure

**Table 2  Deductive comparison between BLE and ESP-NOW protocols.**

| Specification | BLE | ESP-NOW |
|---|---|---|
| Protocol | Standard | Proprietary |
| Communication methodology | Master/Slave | Sender/Receiver |
| Communication mode | Unidirectional | Bidirectional |
| Maximum connected slave nodes | 3 | 7 "up to our experiments" |
| Maximum distance between nodes | 6 meter | 15.5 meter "indoor" 90 meter "outdoor" according to our experiments |
| Power consumption | Very low | Low |

one-way or two-way communication methodologies between the connected ESP32 boards.

### ESP-NOW one-way communication

It is simple to set up one-way communication between ESP32 boards. One-Way communication methodology can be divided into two types: One-to-Many and Many-to-One. In this type of communication methodology, the sent data may be sensor readings or controlling commands (Switching ON and OFF devices, Moving Servo motor, changing RGB color values or other command) (*Random Nerd Tutorials, 2020a*).

As shown in Fig. 4A, one ESP32 board transfer the same or different data to other ESP32 boards in a One-to-Many communication methodology. This setup is suitable for building a remote control system (*Random Nerd Tutorials, 2020a*). As shown in Fig. 4B, one ESP32 board receives data from other ESP32 boards in a Many-to-One communication methodology. This setup is suitable for collecting data from multiple sensor nodes connected to other ESP32 boards (*Random Nerd Tutorials, 2020a*).

### ESP-NOW two-way communication

Two-way communication between ESP32 boards is supported through the ESP-NOW protocol. In this communication style, each board can act as both a sender and a receiver. So ESP32 boards can actually work as a transceiver.

As demonstrated in Fig. 5, the ESP-NOW two-way communication methodology is suitable for creating a mesh network in which many ESP32 boards can transfer data to each other. This methodology can be used to create a network for sharing sensor readings and monitoring system in weather station, construction sites and archaeological sites (*Random Nerd Tutorials, 2020a*).

### Integrating ESP-NOW with Wi-Fi simultaneously

The ESP32 board can be used as a web server in Wi-Fi station mode, Access Point mode, or both. These capabilities enable us to develop a wide range of IoT applications and deploy diverse network architectures (*Vasiljević-Toskić et al., 2019*). As shown in Fig. 6, in some applications, we need to host an ESP32 board as a web server while also integrating it with the ESP-NOW communication protocol (*Random Nerd Tutorials, 2020b*).

Integrating WSNs with the Internet Protocol (IP) to develop Internet of Things (IoT) applications is one of the most important goals for WSNs. IoT systems enables things (a

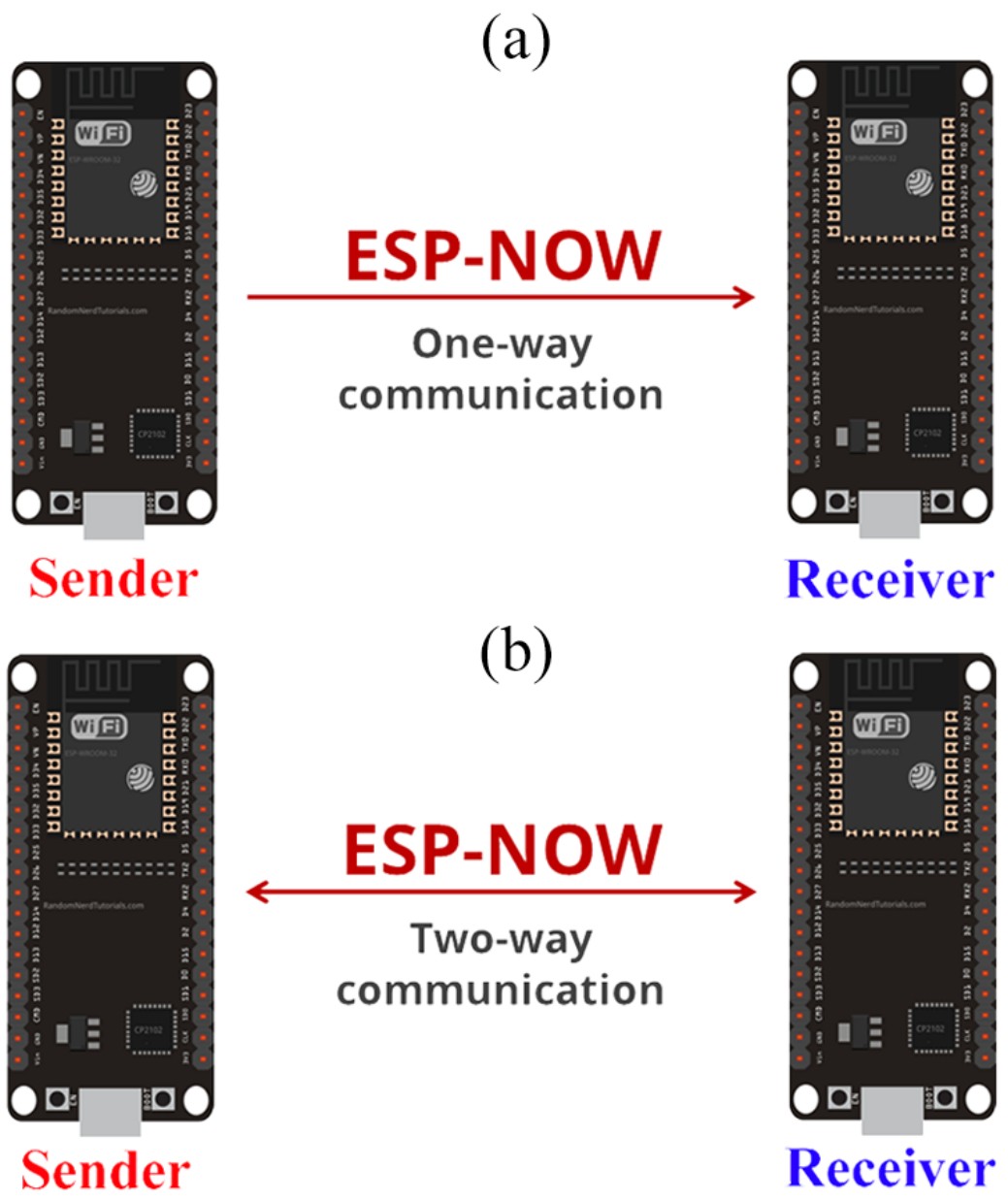

**Figure 3  One-way and two-way communication (adapted from *Random Nerd Tutorials (2020a)*).**

person with a heart monitor implant or a car with built-in sensors to inform the driver when tire pressure is low) to monitor the real-time live objects anytime and everywhere. Integrating ESP-NOW with Wi-Fi in places where Wi-Fi technology is available allows us to build these applications (*Pirbhulal et al., 2017*).

## AN ADAPTIVE SPIDER-MESH TOPOLOGY

The proposed approach acquires its novelty from constructing an adaptive spider mesh topology using the ESP-NOW protocol, which is incorporated into ESP32 devices. Instead

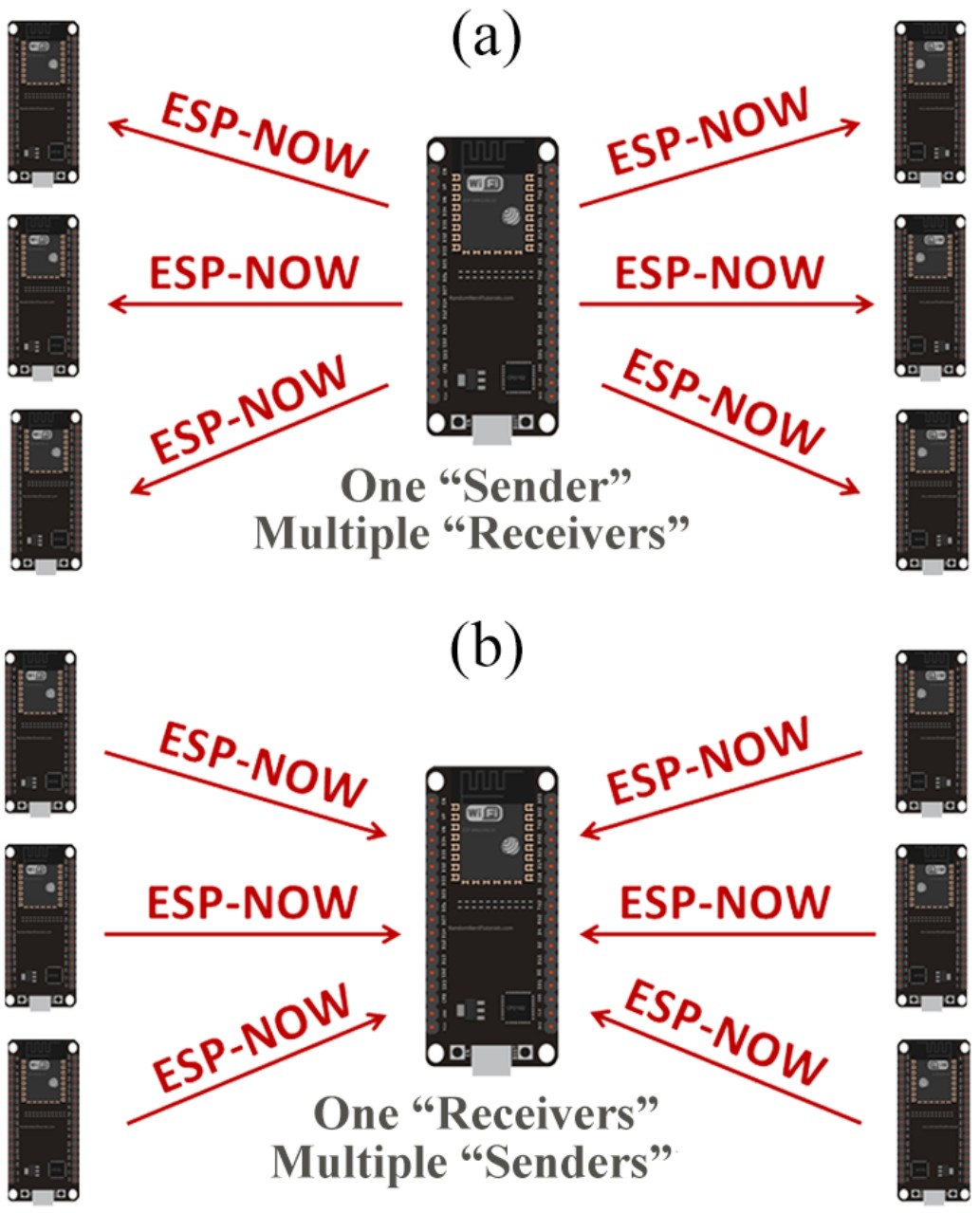

**Figure 4** **ESP32 one-way communication styles (adapted from *Random Nerd Tutorials (2020a)*).**

of using the traditional tree and mesh topologies, the adaptive spider mesh topology is proposed for extending the network coverage. The ESP-NOW protocol can be used as a bi-directional communication protocol that can overcome the BLE protocol's connectivity constraints.

As illustrated in Fig. 7, the adaptive Spider-Mesh topology can have four levels, ranging from 0 to 3. Nodes are labeled with numbers like 1, 2, 3, *etc*. Nodes is this topology are organized into several levels, ranging from 0 to n. So, by increasing the number of levels,

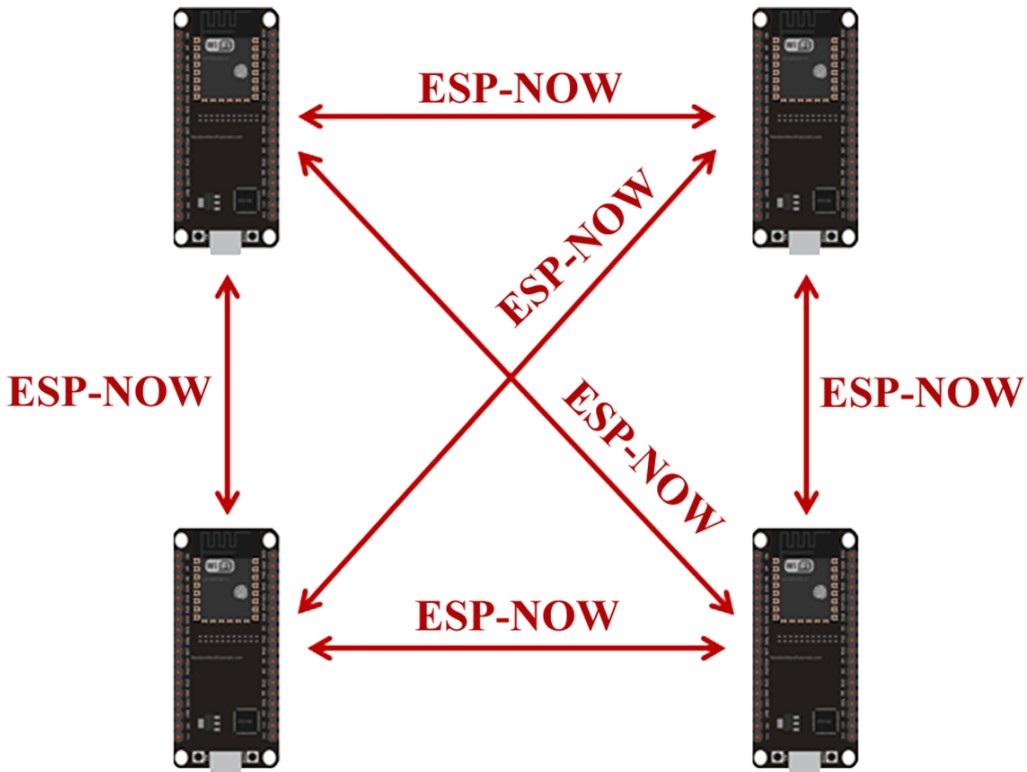

**Figure 5** ESP-NOW two-way communication mesh network (*Random Nerd Tutorials, 2020a*).

the network coverage can be extended.

$$N = 1 + \sum_{i=0}^{n-1} x*(2)^i \quad n \geq 1 \tag{1}$$

The number of nodes can be easily determined using Eq. (1), as shown in Table 3, where N is the total number of nodes that may be connected using the proposed approach, n is the number of levels, and x is the number of nodes in the first level. In the last level, each node is connected to just three other nodes, but in the other levels, each node is connected to five other nodes.

The use of ESP-NOW protocol as a communication technology added many benefits, including the ability to exchange data between ESP32 devices without switching network nodes mode, and the ability to connect one board to seven other boards at once (supporting up to 20 nodes based on recent researches' experiments (*Glória & Sebastião, 2021*)). Using the capabilities of the ESP-NOW protocol, an adaptive Spider-Mesh topology has been proposed for constructing Bi-Directional mesh networks.

The proposed approach studies the possibility of expanding network coverage in locations where traditional Wi-Fi networks or permanent energy sources are difficult to establish, as well as the maximum permissible distance between network nodes using the

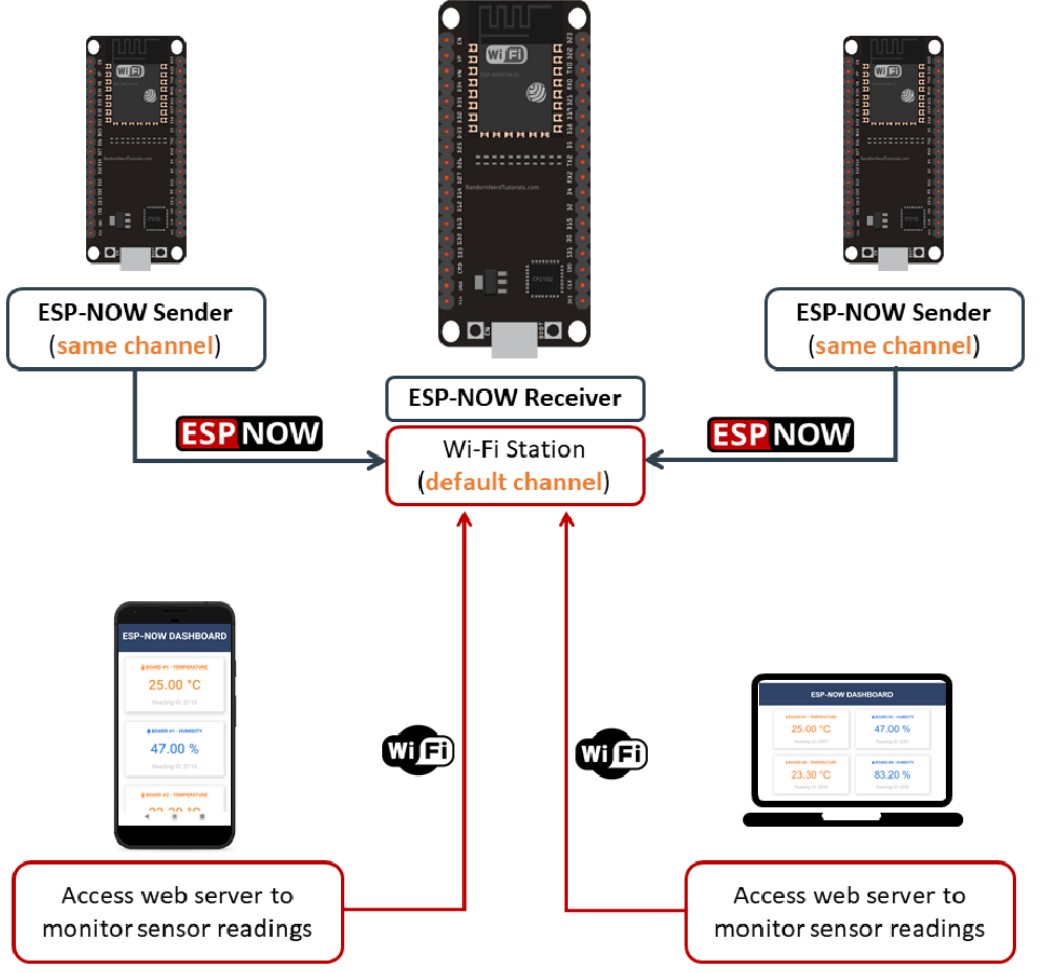

**Figure 6** **Using ESP-NOW and Wi-Fi simultaneously (*Random Nerd Tutorials, 2020b*).**

ESP-NOW protocol inside various indoor and outdoor architectures and the maximum network node lifetime.

## EXPERIMENTAL RESULTS

Two versions of ESP32 development boards were used in the experiments: the standard ESP-32S DEV KIT DOIT board with 30 GPIOs pins and the Wemos D1 R32 UNO ESP32 board. A power supply was also employed, which consisted of Wemos 18650 rechargeable lithium batteries (3.7 v and 4,800 mAh). The Arduino IDE version 1.8.12 has been used to upload code to ESP32 boards in our experiments.

In the first series of experiments, we tested the compatibility between ESP-32S DEV boards and Wemos D1 R32 UNO boards. The two ESP32 boards have been utilized as a transceiver to exchange various data types (up to 250 Mbit/s) from other boards. These experiments revealed that the two versions of ESP32 boards are extremely compatible.

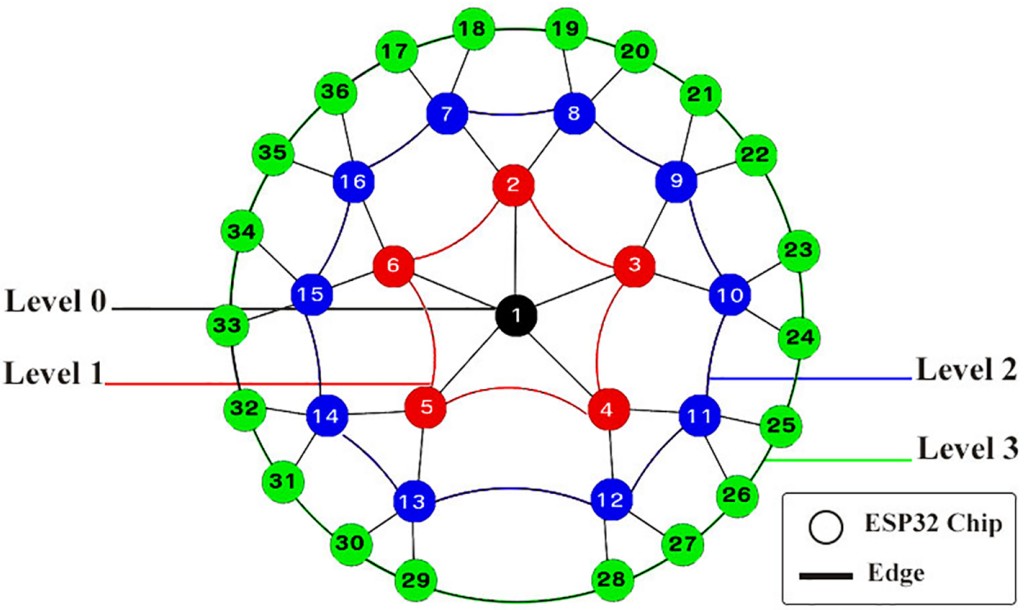

**Figure 7** Four levels of an adaptive Spider mesh topology using ESP32 boards and ESP-NOW protocol.

**Table 3** Number of levels and nodes for the proposed approach.

| Levels | Number of nodes |
|---|---|
| 0 | 1 |
| 1 | $1 + 5 = 6$ |
| 2 | $1 + 5(2)^0 + 5(2)^1 = 16$ |
| 3 | $1 + 5(2)^0 + 5(2)^1 + 5(2)^2 = 36$ |
| n | $N = 1 + \sum_{i=0}^{n-1} x * (2)^i \quad n \geq 1$ |

Exchanging data with the proposed approach through one-way and two-way communication methodologies, to control a group of peripherals based on the received value was the target of the second series of experiments. We were able to ensure the reliability and dependability of the proposed approach in exchanging data. Using the ESP-NOW protocol, we measured the maximum achievable distance between transmitter and receivers under various construction conditions using these experiments on three different indoor architectures and a set of outdoor regions.

The proposed approach was shown to be capable of establishing a simple and fully connected network inside various indoor and outdoor structures in the second series of experiments. As shown in Fig. 8, the maximum distance between network nodes inside various indoor structures is around 15.5 m, while the maximum distance between network nodes for outside environments is roughly 90 m as shown in Fig. 9.

The target of the last series of experiments is to identify the network node lifetime. This target was achieved by measuring the energy consumed in sending and receiving data using

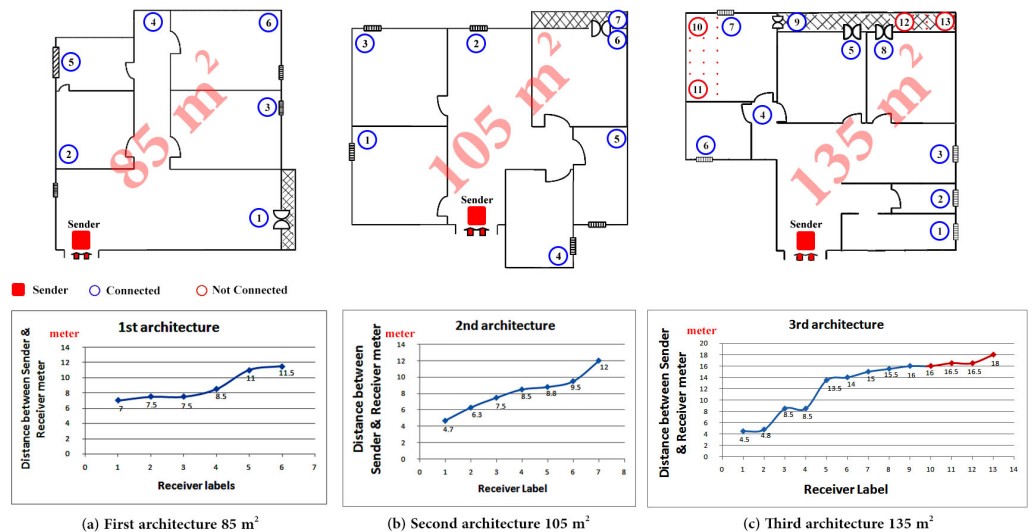

**Figure 8** The distance between sender and receivers inside three different indoor architecture.

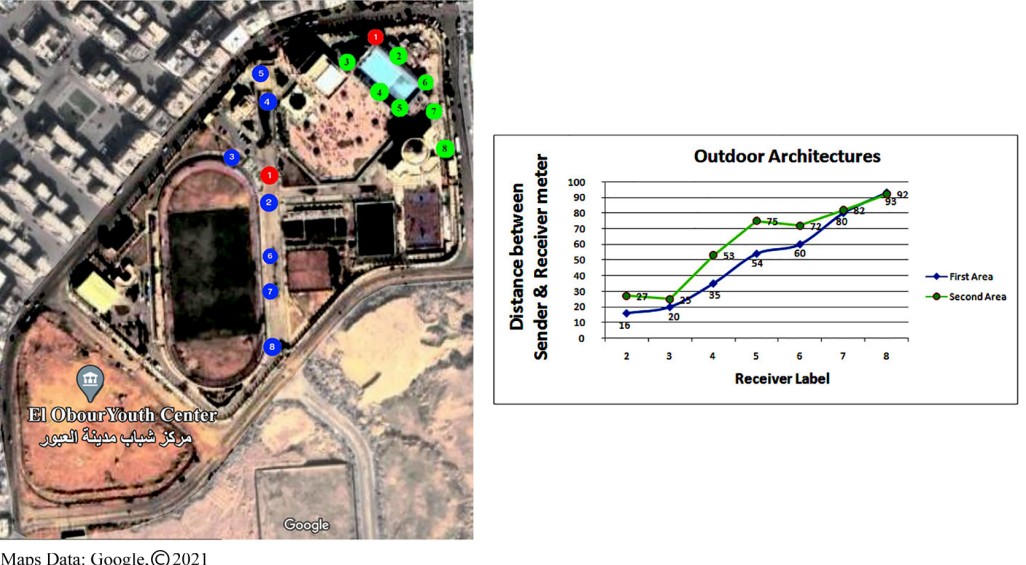

Maps Data: Google,©2021

**Figure 9** The distance between sender and receivers in outdoor architectures.

the Bluetooth Low Energy and ESP-Now protocols.

$$\text{Battery Life}(T) = \frac{\text{Battery Capacity in mAh }(Q)}{\text{Load Current in mA }(I)} \tag{2}$$

The network node lifetime may be easily determined using Eq. (2) (*Farahani, 2011*), where $T$ is the battery lifetime (network node lifetime), measured in hours (h); $Q$ is the battery charge capacity which in our case is 4,800 milliamps per hour (mAh); and $I$ is the

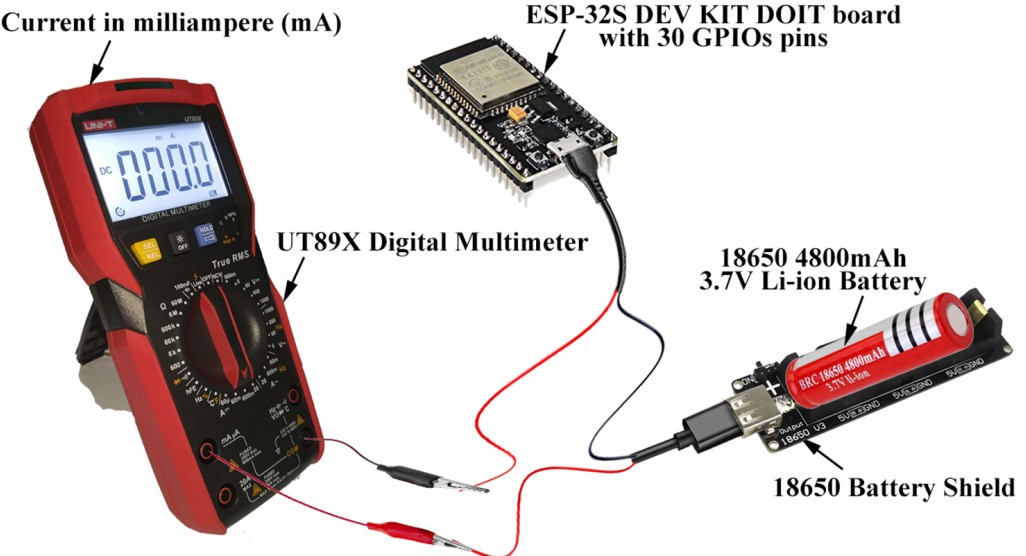

**Figure 10** Using digital multimeter to measure the average load current ($I$).

**Table 4** Experimental results summery using 18650 4800 mAh 3.7V lithium-ion battery.

| Power mode | Protocol | Node state | Power consumption | Node lifetime | Response time |
|---|---|---|---|---|---|
| Active | BLE | Server | 76.4 mA | 62 HRS 49 MIN | 15 ms |
| | BLE | Client | 74 mA | 64 HRS 51 MIN | |
| | ESP-NOW | Sender | 129 mA | 37 HRS 12 MIN | 4 ms |
| | ESP-NOW | Receiver | 121 mA | 39 HRS 40 MIN | |
| Deep-sleep | ULP co-processor is powered on | - | 150 μA | 32000 HRS | - |
| | ULP sensor-monitored pattern | - | 100 μA | 48000 HRS | - |
| | RTC timer +RTC memory | - | 10 μA | 480000 HRS | - |

average current that a load is drawn from it which can be measured in milliamps (mA) as illustrated in Fig. 10.

According to our experiments, the lifetime of a server node using the Bluetooth Low energy protocol is around 62 h (4,800 (mAh)/76.4 (mA), whereas the lifetime of client nodes is roughly 64 h (4,800 (mAh)/74 (mA). The lifetime of a sender node using the ESP-NOW protocol is 37 h (4,800 (mAh)/129 (mA), while the lifetime of a receiver node is 39 h (4,800 (mAh)/121 (mA). Keeping in mind that all previous experimental results are running in active power mode. Table 4 summarizes our experimental results and

**Table 5** Measuring power consumption for the two used communication protocols "ESP-NOW and BLE" on the proposed approach "Based on our experiments".

| Module/ communication protocol | IEEE protocol | Designed for network protocol | $V_{DD}$ (Volt) | $I_{TX}$ (mA) | $I_{RX}$ (mA) | $I_{sleep}$ ($\mu$A) | Max. Bit rate |
|---|---|---|---|---|---|---|---|
| ESP32 (ESP-NOW) | 802.11 | ESP-NOW | 3.3 | 129 | 121 | 10 | 1 (Mb/S) |
| ESP32 (BLE) | 802.15.1 | BLE | 3.3 | 76.4 | 74 | 10 | 1 (Mb/S) - 2 (Mb/S) in Bluetooth 5 |

demonstrates the benefits of using Deep-sleep power mode to save the battery power for thousands of hours.

A comparison between the proposed low power consumption communication protocols in this paper has been summarized in Table 5, while a comparison between an existing short range low power consumption communication protocols has be summarized in Table 6. Figure 11 illustrates the difference between our experimental results and existing researcher results (*Mahmoud & Mohamad, 2016*).

## CONCLUSION AND FUTURE WORK

Recently, researchers are working to build a mesh network with low-power sensors, devices, and protocols in military sites, archaeological sites, smart parking, farmlands, and construction sites. This paper proposes an adaptive low power consumption mesh approach called "An Adaptive Spider-mesh topology". To be able to fairly appraise this study, we must examine network characteristics such as the desired network topology, communication protocol, maximum number of connected nodes simultaneously, communication methods, power consumed, and network node lifetime utilizing the proposed approach.

Although the ESP-NOW protocol is a proprietary protocol that consumes twice as much power of the Bluetooth low energy protocol, the experimental results show that it is an efficient bi-directional communication protocol for developing the proposed approach. Inside various indoor structures, the maximum distance between sender and receiver is roughly 15 m, whereas the maximum distance for outside environments is approximately 90 m. According to our experiments, the transmitter node can simultaneously connect to up to seven receiving nodes. The maximum distance between server and client nodes in the case of BLE protocol cannot exceed 6 m, and the maximum number of connected client nodes to a server node cannot exceed three nodes.

The proposed approach can be used in future work to collect, analyze and monitor unexpected weather conditions that may have severe consequences for construction equipment and materials at construction sites. Choosing the most appropriate energy-efficient routing protocol to the proposed approach is one of the most important challenges we will attempt to address in future research. Machine learning techniques like as Neural Networks, Support Vector Machines, Decision Trees, and other approaches will be integrated with the proposed approach in assisting decision makers or automatically taking decisions in many fields of our daily life. Developing an efficient wireless network

**Table 6 Comparison between existing low power consumption communication protocols (*Mahmoud & Mohamad, 2016*).**

| Module/ communication protocol | IEEE protocol | Designed for network protocol | $V_{DD}$ (Volt) | $I_{TX}$ (mA) | $I_{RX}$ (mA) | $I_{sleep}$ ($\mu$A) | Max. Bit rate |
|---|---|---|---|---|---|---|---|
| ANY900 | 802.15.4 | ZigBee | 3.3 | 33 | 17 | <6 | 250 (Kb/S) |
| MRF24J40MA | 802.15.4 | ZigBee | 3.3 | 23 | 19 | 2 | 250 (Kb/S) |
| RC2400 | 802.15.4 | ZigBee + 6lowpan | 3.3 | 34 | 24 | 1 | 250 (Kb/S) |
| CC2430 | 802.15.4 | ZigBee | 3.3 | 25 | 27 | 0.9 | 250 (Kb/S) |
| deRFmega128-22M00 | 802.15.4 | Zigbee + 6lowpan | 3.3 | 12.7 | 17.6 | <1 | 250 (Kb/S) |
| deRFsam3 23M10-2 | 802.15.4 | ZigBee + 6lowpan | 3.3 | 42 | 40 | <2 | 250 (Kb/S) |
| RN171 | 802.11 b/g | – | 3.3 | 190 | 40 | 4 | 54 (Mb/S) |
| QCA4004 | 802.11 n | – | 3.3 | 250 | 75 | 130 | 10 (Mb/S) |
| GS1011M | 802.11 b | – | 3.3 | 150 | 40 | 150 | 11 (Mb/S) |
| G2M5477 | 802.11 b/g | – | 3.3 | 212 | 37.8 | 4 | 11 (Mb/S) |
| RS9110-N-11-02 | 802.11 b/g/n | – | 3.3 | 19 | 17 | 520 | 11 (Mb/S) |

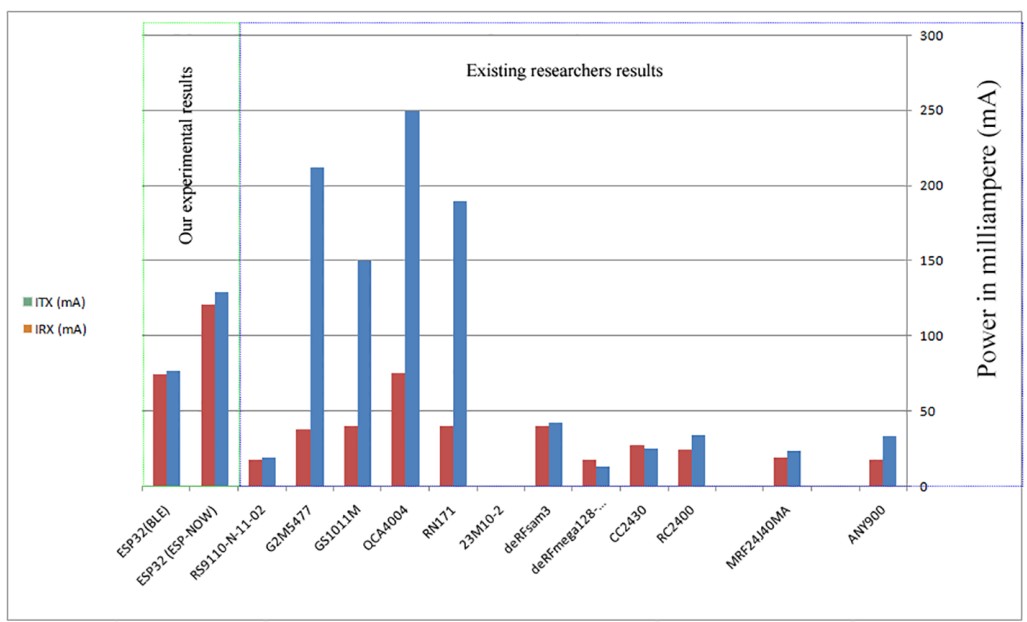

**Figure 11 Comparison chart of existing low power consumption communication protocols with our proposed protocols (combined with *Mahmoud & Mohamad (2016)*).**

solutions using low energy technologies in-body or underwater is an interesting research topic for many researchers.

### Funding
The authors received no funding for this work.

### Competing Interests
Mohamed H. ELgazzar is an Internet of Things Senior Solutions Architect Vodafone International Services, Egypt.

### Author Contributions
- Mostafa Ibrahim Labib conceived and designed the experiments, performed the experiments, performed the computation work, prepared figures and/or tables, authored or reviewed drafts of the paper, and approved the final draft.
- Mohamed ElGazzar analyzed the data, authored or reviewed drafts of the paper, and approved the final draft.
- Atef Ghalwash conceived and designed the experiments, analyzed the data, performed the computation work, authored or reviewed drafts of the paper, and approved the final draft.
- Sarah Nabil AbdulKader conceived and designed the experiments, performed the experiments, performed the computation work, prepared figures and/or tables, authored or reviewed drafts of the paper, and approved the final draft.

### Data Availability
The code files represent part of our experiment in connecting ESP32 devices using one-way, two-way, one-to-many and many-to-one communication methodology and ESP-NOW protocol; as well as using ESP32 in an access point mode and web server mode. Code and data are available in the Supplemental Files.

### Supplemental Information
Supplemental information for this article can be found online at http://dx.doi.org/10.7717/peerj-cs.780#supplemental-information.

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
