# Peer review of "An efficient networking solution for extending and controlling wireless sensor networks using low-energy technologies"

_PeerJ Computer Science, doi:10.7717/peerj-cs.780_

## Round 0.1 · original submission · Major Revisions

Dear Dr. Labib,

Thanks for your submission to PeerJ Computer Science. Could you please respond to the reviewers' comments and we are happy to receive your amended copy.

Reviewer 2 has suggested a specific reference. You may cite it if you believe it is especially relevant. However, I do not expect you to include the citation, and if you do not include it, this will not influence my decision.

Best regards

Abdel-Hamid

Reviewer 1 ·

Basic reporting

no comment

Experimental design

no comment

Validity of the findings

no comment

Additional comments

The authors proposed proposes an approach for constructing and extending Bi-Directional mesh networks using low power consumption technologies inside various indoors and outdoors architectures called "an adaptable Spider - Mesh topology.". The use of ESP-NOW protocol as a communication technology added the advantage of longer communication distance versus a slight increase of consumed power. It provides 15 times longer distance compared to the BLE protocol while consuming only twice as much power.
In my opinion, this work is interesting and should be published after minor revisions.
I consider that the authors should clarify several aspects of the manuscript, for improving the clarity of the presentation. These are listed below.

1. In particular, the term running current is used in Electric Motor Testing, why authors have used the term in the manuscript?
2. The authors can be to consider show their lifetime experimental results in a table.
3. Can the authors measure the time of the response responses time between the different topologies?

·

Basic reporting

This paper proposes an energy-efficient routing protocol for wireless sensor networks. The paper is well-written and easy to follow. Following are my comments:
1. There has been a great amount of work done on developing energy-efficient routing protocols, it would be better to distinguish your work from the existing literature or clearly mention the novel contributions.
2. I wonder if the proposed work can be used in extreme environments such as in-body or underwater {https://repository.kaust.edu.sa/handle/10754/664913, https://ieeexplore.ieee.org/abstract/document/8891506}
3. The results do not show a comparison to existing works to show its effectiveness.

Experimental design

The experimental design is clear.

Validity of the findings

There is no issue with the findings, only comparisons to existing schemes is missing.

---

## Round 0.2 · accepted · Accept

Thanks for your interest in publishing your research in PeerJ Computer Science. It is my pleasure to let you know that your paper has been accepted for publication.

Reviewer 1 ·

Basic reporting

no comment

Experimental design

no comment

Validity of the findings

no comment

Additional comments

I believe that the manuscript is now ready to be published in PeerJ

·

Basic reporting

The authors have addressed all the comments. No further comments from this reviewer.

Experimental design

Experimental design is fine.

Validity of the findings

The findings are valid.

Additional comments

The authors have addressed all the comments. No further comments from this reviewer.